# Mechanisms behind the Immunoregulatory Dialogue between Mesenchymal Stem Cells and Th17 Cells

**DOI:** 10.3390/cells9071660

**Published:** 2020-07-10

**Authors:** Claudia Terraza-Aguirre, Mauricio Campos-Mora, Roberto Elizondo-Vega, Rafael A. Contreras-López, Patricia Luz-Crawford, Christian Jorgensen, Farida Djouad

**Affiliations:** 1IRMB, University of Montpellier, INSERM, F-34090 Montpellier, France; claudia.terraza@inserm.fr (C.T.-A.); rafael.contreras@inserm.fr (R.A.C.-L.); 2IGH, University of Montpellier, CNRS, F-34396 Montpellier, France; maurocamposmora@gmail.com; 3Facultad de Ciencias Biológicas, Departamento de Biología Celular, Laboratorio de Biología Celular, Universidad de Concepción, Concepción 4030000, Chile; relizondo@udec.cl; 4Centro de Investigación Biomédica, Facultad de Medicina, Universidad de Los Andes, Santiago 7620001, Chile; pluz@uandes.cl; 5CHU Montpellier, F-34295 Montpellier, France

**Keywords:** mesenchymal stem cells, Th17 cells, immunoregulation

## Abstract

Mesenchymal stem cells (MSCs) exhibit potent immunoregulatory abilities by interacting with cells of the adaptive and innate immune system. In vitro, MSCs inhibit the differentiation of T cells into T helper 17 (Th17) cells and repress their proliferation. In vivo, the administration of MSCs to treat various experimental inflammatory and autoimmune diseases, such as rheumatoid arthritis, type 1 diabetes, multiple sclerosis, systemic lupus erythematosus, and bowel disease showed promising therapeutic results. These therapeutic properties mediated by MSCs are associated with an attenuated immune response characterized by a reduced frequency of Th17 cells and the generation of regulatory T cells. In this manuscript, we review how MSC and Th17 cells interact, communicate, and exchange information through different ways such as cell-to-cell contact, secretion of soluble factors, and organelle transfer. Moreover, we discuss the consequences of this dynamic dialogue between MSC and Th17 well described by their phenotypic and functional plasticity.

## 1. Introduction

Mesenchymal stem cells (MSCs) are considered a self-renewing cell population present in a wide variety of tissues. Since the first reports by Friedenstein and coworkers during the 1960s and 1970s [1], describing cells with a fibroblast-like morphology and the capacity to proliferate rapidly in discrete colonies, many researchers delved deeper into the study of these cells, eventually establishing that they have the potential to differentiate into chondroblasts, osteoblasts, and adipocytes [2,3,4]. The name “mesenchymal stem cell” was termed in 1991 by Caplan due to their ability to differentiate into different cell lineages [5].

MSCs can be isolated from several adult biological sources including bone marrow, adipose tissue, dental tissue, synovial membrane, peripheral blood, and menstrual blood [6,7,8,9,10,11], as well as perinatal tissues such as umbilical cord, placenta, Wharton jelly, and amniotic fluid [12,13,14,15]. According to the International Society for Cellular Therapy (ISCT), there are three minimum criteria that allow these cells to be identified [16]. Firstly, they must be plastic-adherent when they are kept under standard growing conditions. Secondly, they must express the markers CD74, CD90, and CD105 with a lack of hematopoietic lineage markers such as CD11b, CD14, CD34, or CD45; in addition, they must have a low expression of HLA class II on their surface (<2%). Finally, they must be able to differentiate into cells of mesodermal lineage, more specifically osteoblasts, chondroblasts, and adipocytes, under standard in vitro tissue culture-differentiating conditions [16].

MSCs express a wide variety of chemokine receptors such as CXCR3, CXCR4, or CCR5, allowing them to be recruited from the bone marrow to the circulation and promoting their migration to damaged tissues in pathological conditions [17,18,19]. Additionally, MSCs are able to secrete chemokines such as CXCL12, CX2CL1, CXCL9, CXCL10, or CXCL11, which promote the recruitment of different circulating or resident cell types [19,20].

Independent of their biological sources, MSCs display anti-apoptotic properties which could determine the outcome of diseases and their therapeutic effects [21,22,23,24]. MSCs can also promote the formation of new blood vessels in vitro and in vivo through vasculogenesis or sprouting angiogenesis processes [25,26,27,28,29,30,31]. These cells are reportedly capable of secreting many factors which act in a paracrine way promoting angiogenesis in damaged tissues [32]. Despite this, the MSC secretome may vary depending on the cellular source, hypoxia, and the presence of growth factors and small molecules in the microenvironment [33,34,35]. Additionally, numerous reports suggest that MSCs and the MSC-derived secretome represent a promising alternative for the treatment of fibrosis and damaged tissue cytoprotection [36,37]. 

Due to their potent immunoregulatory and anti-inflammatory properties, MSCs represent one of the most promising cell products to treat autoimmune and inflammatory diseases. Up until 2020, 1042 clinical trials using MSCs were registered in the NIH clinical Trial Database (https://clinicaltrials.gov/), and about 31% (321 of all registered trials) were directed toward treating immune- or inflammation-associated disorders. Autoimmune diseases such as rheumatoid arthritis (RA) are characterized by a predominance of pathogenic immune cells releasing pro-inflammatory cytokines and an alteration of the peripheral immune tolerance due to immune response deficiencies. RA involves T cells and, in particular, CD4^+^ T cells [38,39]. CD4^+^ T cells of RA patients undergo a premature transition from a naïve to a memory phenotype. The resulting memory CD4^+^ T cells are hyper-proliferative and exhibit an enhanced capacity to differentiate into Th1 and Th17 pathogenic T cells [40,41]. In contrast to other T-cell subsets such as Th1 and Th2, Th17 cells and regulatory T cells (Treg) display a high plasticity degree. In RA individuals, the increased frequency of Th17 cells is mediated through either a reduction in the number of Treg or a qualitative defect in their function [42]. Thus, in this context, parallel to the biotherapies that are already widely used in the clinic, MSC-based therapy approaches were investigated for many years [43]. This attractiveness of MSCs to treat autoimmune and inflammatory disorders relies on their capacity to repress the function and the proliferation of pathogenic immune cells while educating regulatory immune cells. We and others demonstrated that MSCs prevent T-cell differentiation into Th17 and induce Treg cells in vitro and in vivo [44,45,46,47,48]. These potent MSC inhibitory/regulatory properties on Th17 cells are mediated through different mechanisms that are tackled in this review.

## 2. Immunoregulatory Properties of MSC

The therapeutic effect of MSCs in several diseases such as graft versus host disease (GvHD), arthritis, multiple sclerosis, fibrosis, systemic lupus erythematosus, or kidney injury, in which the immune system plays a key role, is widely reported [49]. This therapeutic effect is observed mainly due to the ability of MSCs to regulate the activation, proliferation, and function of several subsets of immune cells, from both innate and adaptive compartments (Table 1).

### 2.1. Innate Cells

The differentiation of dendritic cells (DC) from monocyte precursors is affected by MSCs [101]. MSCs not only inhibit the acquisition of the DC mature phenotype, keeping them in an immature state [50,51], but also modulate their functional properties, inhibiting their ability to stimulate lymphocyte proliferation and skewing their cytokine secretion from a pro-inflammatory to an anti-inflammatory profile [52,53,102]. Under inflammatory conditions, MSCs can also modulate the macrophage response, resulting in their differentiation into the type 2 macrophage phenotype (M2) [56,103]. These MSC-modulated M2 macrophages are able to secrete several anti-inflammatory cytokines, possessing tissue remodeling and tolerogenic properties [46]. Moreover, MSCs can downregulate natural killer (NK) cell activation by IL-2 or IL-15, inhibiting NK cell proliferation, cytokine production, and cytotoxicity [63,64,65]. Regarding innate lymphoid cells (ILCs), MSCs stimulate their proliferation, as well as their capacity to release IL-22, which could contribute to immune homeostasis [68].

### 2.2. Adaptive Cells

Adaptive cells such as B and T cells are potently regulated by MSCs. MSCs can suppress B-cell activation by decreasing CD69 and CD86 expression, plasma cell differentiation, and immunoglobulin G (IgG) production [69,70,71]. Moreover, several reports consistently showed that MSCs convert B cells into IL-10-producing B regulatory (Breg) cells which exhibit inhibitory function [72,73,74,104].

Similarly, a compelling body of evidence supports the immunoregulatory effect of MSCs over CD4^+^ T-cell differentiation into the T helper subsets T helper 1 (Th1). MSCs suppress Th1 cell proliferation and the production of pro-inflammatory cytokines such as interferon (IFN)-γ and interleukin (IL)-12, among others [67,76,77]. MSCs can favor T-cell differentiation into the T helper 2 (Th2) subset, increasing IL-4 and IL-5 production [86]. However, MSCs were also shown to inhibit Th2-mediated inflammatory responses in experimental allergy models [87,88,89]. Finally, MSCs promote the differentiation or induction of T cells into Treg cells with immunosuppressive functions. Indeed, these Treg cells are characterized by their capacity to produce anti-inflammatory cytokines such as transforming growth factor-β (TGF-β) and IL-10, along with having tolerogenic properties [59,78,79,90,100].

## 3. Mechanisms behind the MSC Immunosuppressive Effect on Th17 Cells

Among the T-cell subsets that are involved in the immune response, Th17 cells represent a distinct lineage of T lymphocytes located in the lamina propria of the small intestine upon steady state [105]. These pro-inflammatory CD4^+^ T cells express the cytokines IL-17A, IL-17F, and IL-22, as well as the chemokine receptor CCR6 and the RAR-related orphan receptor γT (RORγT), the master transcription factor of both human and murine Th17 cell differentiation and a hallmark of a Th17-specific T-cell signature [41]. In response to an inadequate or deficient T-cell immune regulation, Th17 cells participate in the development of several autoimmune diseases, and several studies described Th17 cells as major players in the pathogenesis of many inflammatory and autoimmune diseases, such as rheumatoid arthritis, type 1 diabetes, multiple sclerosis, systemic lupus erythematosus, and bowel disease [106,107]. Th17 and Treg cells are highly plastic versatile cells that can adopt different phenotypes and functions when exposed to physiological or pathological conditions [108]. Hence, the balance between Treg cells and Th17 cells (Treg/Th17 ratio) often shapes the outcome of immune responses (immunosuppression versus inflammation) and, thus, represents a promising immune target to determine the effectiveness of immune therapy [108,109].

Due to their immunomodulatory and protective properties, MSCs are studied for their therapeutic potential in inflammatory and autoimmune diseases. Although the mechanisms via which MSCs inhibit the response of different immune cell populations, particularly the Th17 subsets, are not fully elucidated; to date, several processes were reported. Indeed, MSC immunoregulatory properties are mediated through the production of soluble factors, mechanisms dependent on cell-to-cell contact, mitochondria transfers, or the production of extracellular vesicles containing microRNAs (miRNAs) among other molecules (Figure 1).

### 3.1. Soluble Factors

The production of soluble factors by MSCs that mediate the immune response is widely studied.

Prostaglandin E2 (PGE2) is a lipid molecule capable of exerting its immunomodulatory effect in several immune cell populations, including Th1 and Th17 cell subsets [110]. However, depending on the maturation state of the different lymphocyte subsets, PGE2 can display a dual role. In mature Th1 cells, PGE2 inhibits their proliferation and the production of IFN-γ, whereas, during Th1 polarization, there is an increase in the secretion of IFN-γ [111]. In contrast, the opposite effect was observed in Th17 cells. During Th17 differentiation from naïve T cells in vitro, PGE2 increases the expression of IL-23R and IL-1R via the activation of PGE2 receptor 2 (EP2)- and receptor 4 (EP4)-dependent signaling pathways, along with the increase in cyclic AMP pathways [112]. However, when memory CD4^+^ T cells are exposed to Th17-skewing conditions, MSC-derived PGE2 inhibits IL-17A secretion via an EP4-mediated, contact-dependent mechanism [113,114]. This differential response to secreted PGE2 in Th1 and Th17 cell subsets might be mediated partially by the interaction with other cell types, such as myeloid cells, since selective removal of myeloid cells from a co-culture of MSCs and CD4^+^ T cells impairs Th17 immunosuppression mediated by PGE2 [48]. Nevertheless, this dual effect of PGE2 requires to be fully elucidated. Recent studies with PBMCs obtained from RA patients (co-cultivated with adipose tissue-derived MSCs) showed an improvement in the Treg/Th17 ratio. as well as an increase in the production of TGF-β and a decrease in the IL-17 levels [115]. The authors proposed that this effect was mediated by a series of factors secreted by MSCs, including PGE2 [48]. Injection of umbilical cord-derived MSCs in patients with active systemic lupus erythematosus (refractory to conventional therapies) was associated with an increased frequency of Foxp3^+^ Treg cells one week post MSC injection and up to three months later with an additional decrease in frequency of Th17 cells from three to 12 months post injection [96]. In this study, the authors evaluated the potential in vitro mechanisms behind this effect and concluded that the upregulation of Treg cells and downregulation of Th17 cells was mediated by PGE2 and TGF-β [96].

Several cytokines were shown to be involved in the immunosuppressive function of MSC. The anti-inflammatory cytokine IL-10 is secreted by MSCs, and previous studies reported that IL-10 secreted by MSCs inhibits the differentiation of naïve CD4^+^ T cells into Th17 cells in vitro, by downregulating RORγT and impairing the signaling pathways related to RORγT activation [116]. Despite this, more recent in vitro studies showed that MSC does not impair IL-23R expression on Th17 cells but strongly modulates the Th17 cell secretome [117]. MSCs themselves can secrete TGF-β and, thus, induce the differentiation of conventional CD4^+^ CD25^−^ T cells into Foxp3^+^ Treg cells in vitro [118]. In addition to the canonical immunosuppressive pathways triggered by the binding of MSC-derived TGF-β on T cells, it was shown that the TGF-β produced by MSCs also modulates the function of some caspases such as the caspase 8, critical for T-cell activation in a renal xenograft mice model [119]. In addition, TGF-β, produced by MSC along with IL-10, suppresses Th17 differentiation mediated by dendritic cells [54,102]. Despite being described as a pro-inflammatory cytokine, IL-6 is massively produced by MSCs. This cytokine is essential for MSCs to produce PGE2 in vitro and to decrease inflammation in the murine model of collagen-induced arthritis (CIA) [120]. Accordingly, IL-6 released by MSCs contributes to decreased Th17 cell frequency and inhibits the production of IL-17 and IL-22 in vitro released by PBMCs from RA patients. This resulted in the increase of both Foxp3^+^ Treg cells and TGF-β produced by these PBMCs [115]. Moreover, MSCs secrete IL-1 receptor agonist (IL-1Ra) which inhibits IL-1α and IL-1β signaling and, thus, impacts the activation and function of several immune populations such as macrophages, DCs, and lymphocytes [121]. Recent studies showed that IL-1Ra produced by MSCs inhibits Th17 polarization in vitro. In vivo, in an IL-1ra knockout (KO) mice experimental arthritis model, MSC injection reduced the severity of arthritis while increasing the Treg/Th17 ratio as compared to the control group [77].

Another key mediator of MSC immunoregulatory properties, specific for human MSCs, is indoleamine 2, 3-dioxygenase (IDO), which promotes the degradation of tryptophan into kynurenine (a toxic metabolite), thus inhibiting T-cell proliferation or inducing apoptosis [122,123]. Kynurenine inhibits Th17 cells by suppressing the mammalian target of the rapamycin (mTOR) pathway and through tryptophan deprivation followed by the induction of general control non-depressible 2 (GCN2), a stress response kinase [124,125,126]. The infusion of MSC (or their conditioned media) in a model of carbon tetrahydrochloride-induced murine liver fibrosis attenuated fibrosis progression. These results were accompanied by higher serum levels of IDO, IL-10, and kynurenine and lower levels of IL-17. Moreover, the frequency of CD4^+^ IL-10^+^ was increased but the number of Il-17^+^ Th17 cells was reduced. These results were completely reversed in the presence of an IDO inhibitor [127]. 

Over the last few years, the enhancement of MSC immunoregulatory properties was intensively investigated through the development of different protocols of MSC pretreatment. For example, the pretreatment of adipose tissue-derived MSCs with metformin revealed an enhanced IDO expression in MSC in a STAT1-dependent manner [75]. The infusion of these cells in an experimental model of lupus significantly increased the Treg/Th17 ratio, as well as the production of anti-double stranded DNA (dsDNA) IgG antibodies, reducing inflammation [75].

Heme oxygenase-1 (HO-1) is an inducible enzyme with anti-inflammatory and immunoregulatory properties, which catalyzes the conversion of heme into carbon monoxide (CO), biliverdin, and Fe (2^+^) [128]. Biliverdin production protects cells against oxidative damage, while CO has a similar effect to nitric oxide (NO) by stimulating guanylyl cyclase and increasing intracellular levels of cGMP, reducing leukocyte adhesion, decreasing apoptosis, and downregulating the production of some pro-inflammatory cytokines [128]. The infusion of modified bone marrow-derived MSCs expressing higher levels of HO-1 ameliorated the severity of GvHD in a murine model by regulating the Treg/Th17 balance in lymph nodes and the spleen [97]. More recently, the overexpression of HO-1 by MSCs was shown to inhibit NK cells, reduce the Th1/Th2 balance, and induce Th17 into Treg in vitro [129]. When these modified MSC were tested in vivo in a reduced-size liver transplant rejection model, animals showed a lower transplant rejection rate and pro-inflammatory cytokine levels along with higher anti-inflammatory cytokines levels and number of peripheral Treg [129]. Together, these results suggest that the overexpression of HO-1 or activity in MSCs could be useful to improve MSC efficacy in pre-clinical and clinical applications [130].

Unlike IDO and HO-1, inducible nitric oxide synthase (iNOS) is an enzyme produced exclusively by murine MSC. iNOS was shown to exert an inhibitory effect on different subsets of T lymphocytes by suppressing their proliferation through the production of NO [131]. At high concentrations, NO inhibits TCR-induced T-cell proliferation and cytokine production [132]. NO produced by murine MSCs suppresses the phosphorylation of the signal transducer and activator of transcription 5 (STAT-5) and inhibits CD4^+^ T-cell proliferation [133]. MSC pretreatment with IFN-γ and other pro-inflammatory cytokines, such as tumor necrosis factor-α (TNF-α) or IL-1β, increases their capacity to produce NO and other chemokines, enhancing their immunomodulatory properties [133]. A recent study supported the importance of MSC licensing prior to their administration in vivo. Indeed, when MSCs were primed with TNF-⍺ and IL-1β, they suppressed lymphocyte proliferation through NO production. This effect was demonstrated in vivo in a rat cornea transplant model [59]. The treatment with licensed MSCs significantly improved graft survival in comparison to non-licensed MSCs, and this was evidenced by an increasing number of Treg cells in draining lymph nodes, spleen, and lungs, in addition to lower levels of pro-inflammatory cytokines [59]. In another recent study, the immunomodulatory effect of MSCs was evaluated in a model of facial nerve injury in rats [134]. When the facial nerve was harvested and co-cultured in vitro with bone marrow-derived MSCs, an upregulation of iNOS (among other immunosuppressive factors) was observed. The local infusion of MSC after nerve face injury led to a decrease in the frequency of Th17 cells in lymph nodes and an antiapoptotic effect in facial motor neurons [134].

Some studies showed that the production of hepatocyte growth factor (HGF) by MSCs can partially modulate the Treg/Th17 balance. An in vitro study using MSC co-cultured in transwell with CD4^+^ T cells activated with LPS proposed that HGF is necessary for an increased number of CD4^+^ Foxp3^+^ cells and a downregulated number of CD4^+^ RORγt^+^ cells [135]. In an in vivo model of bronchiolitis obliterans in mice, the infusion of umbilical cord-derived MSCs overexpressing HGF improved the Th1/Th2 ratio, increased the number of Treg cells, and decreased the number of Th17 cells in the spleen; these results were accompanied by decreased IFN-γ levels and increased IL-4 and IL-10 levels in serum [136]. While the expression of c-Met (HGF receptor) on Th17 is not yet reported, it is known that this receptor is expressed in immune cells with antigen-presenting capacities, such as DCs. Therefore, an indirect effect of HGF on Th17, with DCs as intermediaries, cannot be discarded [137].

### 3.2. Cell-to-Cell Contact

One molecule considered crucial for cell-to-cell contact-based inhibition is programed death 1 (PD-1). This molecule is necessary for the downregulation of immune responses by inducing apoptosis of target cells when it binds with its ligand (PD-L1) [138]. It is expressed in various cell types, including B and T lymphocytes [138]. When MSCs are co-cultured with mature Th17 lymphocytes, expression of PD-L1 is significantly increased on the surface of MSCs, promoting the inhibition of Th17 cell proliferation. This latter effect is reversed by using PD-L1-neutralizing antibodies [44]. It was shown that palatine tonsil-derived MSCs are capable of inhibiting Th17 differentiation in vitro through the PD-L1/PD-1 axis, and this effect is enhanced by the secretion of IFN-β by MSCs [139]. When these MSCs were locally administered in vivo in an imiquimod-induced psoriatic skin inflammation model, the disease symptoms were significantly decreased mainly by affecting Th17 response in a PD-L1-dependent manner [139].

Another important mechanism that depends on cell-to-cell contact relies on the Fas/FasL axis. The systemic infusion of bone marrow-derived MSC ameliorated the disease phenotypes in the in vivo models of fibrillin-1 mutated systemic sclerosis (SS) and dextran-sulfate-sodium-induced experimental colitis. This effect was mediated by T-cell recruitment through the secretion of monocyte chemotactic protein 1 (MCP-1) by MSC. In addition, their posterior apoptosis was triggered by the Fas/FasL union. This apoptotic effect promoted the recruitment of macrophages, which secreted high levels of TGF-β, promoting the upregulation of Foxp3^+^ regulatory T cells [58]. This same effect was also reported in Th17 cells, using gingiva-derived MSCs in a colitis mice model [98,99].

Adhesive molecules such as vascular cell adhesion protein (VCAM) and intercellular adhesion molecule (ICAM) were also described to play an important role in the immunosuppressive properties of MSC. These adhesive molecules promote proximity between MSCs and lymphocytes, improving the regulatory effect promoted by the secretion of soluble factors, such as NO, by MSCs [140,141].

### 3.3. Extracellular Vesicles

Most somatic cells release extracellular vesicles (EVs), which are membrane vesicles involved in cell-to-cell communication and several physiological processes that comprise different vesicle subsets which share features such as size and biochemical composition [142]. While exosomes (30–150 nm diameter) originate from endosomal compartments derived from the plasma membrane, microvesicles (100–1000 nm diameter) originate mainly from the plasma membrane while apoptotic bodies (50–4000 nm) emerge from the fragments of dying cells. The EV composition largely reflects that of the parenting cell and, along with EV-enriched proteins (e.g., Alix, Tsg101, CD63, HSP90), MSC-derived EVs also reportedly contain MSC markers such as CD44, CD90, and CD105 [143]. However, the immunosuppressive functions of MSC-derived EVs are predominantly attributed to other proteins, messenger RNAs (mRNAs), and non-coding RNAs, such as miRNAs that modulate gene expression by affecting the translation of other specific mRNAs [144,145]. Therefore, MSC-derived EVs are able to modulate the function of several immune populations such as inhibiting B-cell proliferation and plasmablast differentiation, modulating macrophage polarization, inhibiting DC maturation, impairing neutrophil mobilization, and suppressing NK cell and T lymphocyte proliferation, among others [54,60,61,66,81,82,146,147]. 

Although the mechanisms of MSC-derived EV regulation on Th17 cells are not yet fully understood, some reports indicated that these EVs downregulate the Th17 polarization of in vitro activated CD4^+^ T cells from PBMCs of healthy donors [94,95]. An additional study found that, among PBMCs from type 1 diabetes (T1D) patients, in vitro activation of islet antigen-specific T cells was inhibited by MSC EVs, decreasing the frequency of Th17 cells and IL-17 levels via a mechanism involving TGF-β and PGE2. These EVs also showed promising results as a treatment to control inflammatory immune responses in several animal models [147]. Studying the therapeutic potential of MSC EVs in Theiler’s murine encephalomyelitis virus (TMEV)-induced demyelinating disease (a progressive model of multiple sclerosis), it was observed that MSC EV treatment decreased Th17-derived IL-17 serum levels and brain immune infiltration, which was associated with lower production of several microglial-derived pro-inflammatory cytokines [83]. Although EV treatment reduced brain atrophy and increased remyelination in TMEV-infected mice, the exact mechanism via which these MSC-EVs might contribute to ameliorate this disease remains to be elucidated [83,148]. 

However, other studies attributed to miR-21 the protective effect of MSC EVs inhibiting Th17 cell-driven inflammation, mainly by reducing renal DC maturation and cytokine secretion, using an ischemia/reperfusion injury model of acute kidney disease [149,150,151]. In addition, using the model of experimental autoimmune uveitis (EAU), it was shown that intravenous or periocular administration of MSC EVs decreased the number and frequency of IL-17-producing Th17 cells in immunized mouse’s eyes, which was confirmed using a T1D mice model showing decreased transcript levels of the DC-derived Th1- and Th17-related cytokines IL-1β, IL-6, and IL-12 [84]. Altogether, these reports implicate that EVs derived from MSCs exert a crucial role controlling inflammatory responses and they have significant potential for treating Th17-mediated autoimmune diseases. More research is needed to elucidate the participation of additional components and factors present in these EVs.

### 3.4. Transfer of Mitochondria

Mitochondrial function is necessary for oxidative phosphorylation, ATP generation, and several metabolic pathways in somatic cells, while its dysfunction leads to ROS overproduction and oxidative damage to cells [152]. Remarkably, it was observed that MSCs exert a cytoprotective function to damaged cells via mitochondrial transfer, leading to improved survival and modulation of cell proliferation, as well as signaling both in vitro and in vivo [153,154,155]. 

So far, MSCs were shown to transfer their mitochondrial load via contact-dependent mechanisms such as gap junctions, transient cell fusion, and tunneling nanotubes (TNT), in addition to contact-independent mechanisms such as mitochondria-containing EVs and isolated mitochondria transfer [57,156,157,158]. This mitochondrial transfer from MSCs exhibited cytoprotective effects in kidney injury, myocardial damage, injured retinal ganglion, pulmonary alveoli, and damaged cornea, among others [159,160,161]. Nevertheless, the understanding of the underlying molecular mechanisms of mitochondrial transfer and recipient cell repercussions remains unclear.

To date, there were very few studies focusing on the mitochondrial transfer from MSC to immune cells. As described by other cell types, MSCs were shown to transfer mitochondria to macrophages through both TNT- and EV-mediated mechanisms, which increased the phagocytic capacity of macrophages in a mouse model of *Escherichia coli* pneumonia, or modulated macrophages into an anti-inflammatory M2 phenotype, reducing lung inflammation and injury in mice [57,62]. Additionally, the artificial transfer of MSC-derived mitochondria reportedly induced Treg cell differentiation from activated human CD4^+^ T cells, and these pre-treated T cells with MSC mitochondria reduced leukocyte tissue infiltration and improved animal survival in a GvHD mouse model [85]. However, how naturally occurring mitochondrial transfer impacts T-cell activation and function still remains insufficiently described. 

Previous reports from our group showed that MSCs exert immunosuppression to pathogenic Th17 cells in the context of rheumatoid arthritis (RA) [41,162]; thus, we aimed to investigate whether MSCs modulated the inflammatory environment in RA patient joints through mitochondrial transfer to T cells. When we cultured MSC with ex vivo expanded human Th17 cells, we observed a contact-dependent mitochondrial transfer that occurred as soon as four hours after co-culture [47]. We observed a decrease in IL-17 production of these modulated Th17 cells, and a portion of these cells interconverted into Foxp3^+^ Treg cells. Moreover, oxidative phosphorylation and oxygen consumption were increased in these MSC-treated Th17 cells, suggesting a metabolic switching associated with MSC immunomodulation and Th17–Treg interconversion [47]. Considering that MSCs are present in the synovium during RA onset, we wanted to reveal whether this mitochondrial transfer to CD4^+^ T cells was altered in MSCs from RA patients (RA-MSCs) compared to MSCs from healthy donors, eventually finding that mitochondrial transfer capacity of RA-MSCs was significantly lower compared to healthy MSCs [47]. Altogether, these results suggested that impaired mitochondrial transfer from MSC in the context of RA pathogenesis (and maybe in other autoimmune diseases) could contribute to inflammation and joint damage, worsening the outcome of the disease. However, additional studies are definitely needed to clarify the molecular mechanisms involved in this transfer and the contribution of metabolic switching in the immune function and phenotype of modulated T cells during RA.

## 4. MSC Enhancement to Improve Their Therapeutic Potential

Stimulating MSCs with biological, chemical, or physical factors was proven to be an efficient strategy to enhance their therapeutic function [163]. Several studies demonstrated that the activation of MSC with pro-inflammatory cytokines, as well as with growth factors, induces their multiple immunosuppressive mechanisms. For example, the pre-treatment of MSCs with IFN-γ prior to being co-cultured with activated lymphocytes enhanced their capacity to decrease the production of IFN-γ and TNF-α, increased the secretion of IL-6 and IL-10, increased the frequency of CD4^+^ CD25^+^ CD127^dim/−^ regulatory T cells, and decreased the frequency of Th17 cells [164]. Moreover, IL-1β-primed MSCs were shown to upregulate the expression of genes related to several biological processes linked to the NF-κB pathway [165], and the infusion of these cells in a murine colitis model led to the polarization of peritoneal M2 macrophages, increased frequencies of Treg cells, and decreased the percentage of Th17 cells in the spleen and mesenteric lymph nodes [166]. Considering the interaction between Th17 and MSC, it was described that IL-17A, the main cytokine produced by Th17 cells, enhances the immunomodulatory properties of murine MSC, both in vitro and in vivo [167,168]. This effect depends on the expression of IL-17 receptor A (IL17RA) on the MSC surface, which is involved in the surface levels of VCAM1, ICAM1, and PD-L1, along with iNOS expression [167,168]. Moreover, one report showed that human MSCs treated with IL-17A exhibited a higher in vitro T-cell suppression of proliferation, a lower proinflammatory cytokine production, and a higher induction of Treg cells with no associated upregulation of major histocompatibility complex (MHC) class I and II compared to MSCs treated with IFN-γ [169]. However, some disadvantages were reported, including an increased immunogenicity of MSCs after IFN-γ stimulation, the elevated costs of recombinant cytokines, and variability in the response of MSCs from different sources [163]. 

Three-dimensional (3D) spheroid culture conditions were also shown to enhance MSC immunoregulatory functions. Indeed, human MSCs significantly increased their capacity to produce and release suppressive factors such as IDO when cultured as 3D aggregates [170]. Since oxygen availability in the BM compartment is quite limited, going as low as 1% [171], several studies already demonstrated that MSCs cultured under hypoxic conditions had increased production of soluble bioactive factors, higher angiogenic potential, and immunomodulatory activity [172]. MSCs cultured under hypoxic conditions induced the production of hypoxia-induced factor-1 alpha (HIF1α) expression, which is associated with the production of these multiple mechanisms mentioned, increasing the suppressive potential on Th1 and Th17 cells [173,174].

The peroxisome proliferator activator receptor β/δ (PPARβ/δ) was suggested to play a critical role in the control of the therapeutic potential of MSCs. PPARβ/δ is a transcription factor from the nuclear receptor superfamily that exhibits diverse biological functions like inflammation regulation through the inhibition of NF-κB signaling, which was described in adipocytes and macrophages [175,176]. As previously mentioned, the stimulation of NF-κB signaling with pro-inflammatory cytokines activates the immunosuppressive effect of MSCs. Since PPARβ/δ is a negative regulator of NF-κB activity, the inhibition of PPARβ/δ, either chemically or genetically on MSCs, enhanced their therapeutic potential both in vitro and in vivo in an experimental model of arthritis, diminishing the frequency of Th17 cells. This enhanced immunosuppressive potential of MSCs inhibited through PPARβ/δ was associated with a significant increase of NF-κB activity [177]. Several other strategies were proposed to enhance the anti-arthritic and immunoregulatory properties of MSCs on Th17 cells including the expression of glucocorticoid-induced leucine zipper (Gilz) by MSC, which induces Treg differentiation [45,178]. However, further research has to be completed in order to elucidate the molecular mechanisms associated and turn it into a future clinical tool.

Although these approaches showed promising results in vitro and in pre-clinical animal studies using both murine and human MSCs, they are pending elucidation to see if these strategies are translatable to clinical studies. For example, the treatment of MSCs with GSK3787, a selective and irreversible antagonist of PPARβ/δ that provides MSCs with a potent immunosuppressive effect, seems promissory for the treatment of patients with autoimmune diseases such as RA [177]. However, despite this drug being described as non-toxic and non-bioaccumulative in tissues [179], it is not currently approved by the United States (US) Food and Drug Administration (FDA), which maintains PPARβ/δ as an attractive therapeutic target for MSC immunosuppressive function; however, it will require additional research to find new drugs. In 2007, a phase Ia clinical trial aimed to prove the safety of a PPARβ/δ agonist was successful and a phase Ib trial was started [180]; therefore, future studies cannot be discarded to test the safety of PPARβ/δ antagonists. This is one of the major aspects to consider in order to apply further studies for clinical therapies. 

## 5. Conclusions

MSCs and Th17 cells are two plastic and versatile cells that interact both in physiological and in pathological conditions with different phenotype and functions. MSCs are not immunoregulatory per se. To acquire an immunoregulatory phenotype and function, and to regulate immune responses, MSCs need to be activated by factors released by pro-inflammatory immune cells such as Th17 cells. Once they acquire an immunoregulatory status, MSCs repress the differentiation program of Th17 cells, which is accompanied by T cells producing IL-10 and expressing Foxp3. This bi-directional immunoregulatory dialogue is mediated through different mechanisms that include the release of soluble factors and extracellular vesicles, a reservoir of proteins, lipids, mRNAs, and miRNAs, cell-to-cell contact, and the transfer of organelles (Figure 1). However, considering the mechanisms that govern the dialogue between MSCs and Th17 cells and the plasticity of the two cell types (in addition to the poor survival rate of MSCs upon in vivo injection), the immunoregulatory properties exerted by MSCs and Th17 cells can be effective only for a restricted period of time. Therefore, efforts to prolong MSC persistence in a pathological environment should be attempted in order to extend the exposure of Th17-infiltrated diseased tissue to MSC-derived immunoregulatory, cytoprotective, and therapeutic products.

## Figures and Tables

**Figure 1 cells-09-01660-f001:**
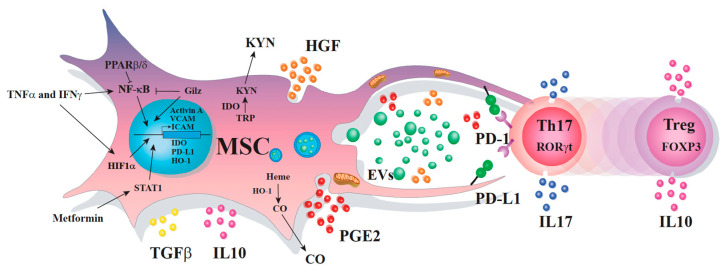
The multifaceted immunoregulatory dialogue between MSCs and Th17 cells. MSCs and Th17 cells communicate through many different ways that include the release of soluble factors, the transfer of mitochondria, the production of extracellular vesicles containing various molecules, and cell-to-cell contact. When MSCs and Th17 cells are near one another, they dialogue through the release of messengers including PD-L1, HGF, and PGE2 that are highly expressed when MSCs are co-cultured with Th17 cells. Moreover, MSCs (nearby Th17 cells) start to express high levels of IDO and HO-1, releasing some EV-containing immunoregulatory factors and transferring their mitochondrial via transient cell fusion and tunneling nanotubes. In return, Th17 cells start to express high levels of PD-1, exhibiting a reduced production of IL-17 and expression of RORγT, while increasing the capacity to release IL-10 and to express Foxp3. This type of signaling, in which MSCs and Th17 cells communicate over relatively short distances, regulates the phenotype and function of both cells.

**Table 1 cells-09-01660-t001:** Mechanisms of mesenchymal stem cell (MSC) immunosuppression on immune cells. Recapitulation of recent reports describing the known mechanisms via which MSCs exert suppressive function to different immune populations.

Target Cell	Involved Mechanism	Observed Effect	References
**Innate Cells**			
DCs	TGF-β, HGF, EVs	↓MHC-II, CD86, CD40; ↑phagocytic function; ↓IL-6, IL-12, TNF-α, IFN-α; ↑IL-10, TGF-β	[50,51,52,53,54]
Macrophages	TGF-β, iNOS, mitochondrial transfer, EVs	↓CD86; ↑phagocytic function, ↓IL-6, IL-8, TNF-α, IL-1β; ↑IL-10, TGF-β; ↑M2 type polarization	[55,56,57,58,59,60,61,62]
NK cells	IDO, PGE2, EVs	↓IFN-γ; ↓cytotoxic activity, proliferation	[63,64,65,66]
Neutrophils	TGF-β, EVs	↓CRAMP and MPO messenger RNA (mRNA), ↓IL-17	[52,57,61,67]
ILC	PGE2, IL-7	↑IL-22; ↑proliferation	[68]
**Adaptive Cells**			
B cells	PD-1, IDO, TGF-β, EVs	↓IgG production; ↓CD69, CD83, CD86; ↓IL-4 mRNA; ↓proliferation, plasmablast differentiation; ↑IL-10	[66,69,70,71,72,73,74,75]
Th1 cells	PD-1, IDO, Fas, IL-6, TGF-β, IL-1Ra, EVs, mitochondrial transfer	↓IFN-γ, IL-1β, TNF-α; ↑apoptosis; ↓proliferation; differentiation	[52,58,67,76,77,78,79,80,81,82,83,84,85]
Th2 cells	IL-6, EVs	↑differentiation; ↑IL-4, IL5; ↓IL-4, IL-5, IL-13	[67,76,86,87,88,89]
Th17 cells	PGE2, Fas, IDO, IL-6, TGF-β, IL-1Ra, EVs, mitochondrial transfer	↓IL-17, IL-22; ↑apoptosis; ↓proliferation; differentiation; ↑interconversion to Treg cells.	[54,58,67,75,76,77,78,79,81,82,83,84,90,91,92,93,94,95,96,97,98,99]
Regulatory T (Treg) cells	PGE2, Fas, IDO, IL-6, iNOS, TGF-β, IL-1Ra, EVs, mitochondrial transfer	↑PD-1, ↑IL-10, TGF-β; ↑proliferation; differentiation; ↑conversion from Th17 cells.	[54,58,59,66,67,75,76,77,78,79,80,81,82,85,90,91,92,93,94,96,97,98,99,100]

CRAMP: cathelicidin-related antimicrobial peptide; DCs: dendritic cells; EVs: extracellular vesicles; HGF: hepatocyte growth factor; IDO: indoleamine 2,3-dioxygenase; IFN: interferon; IgG: immunoglobulin G; IL: interleukin; ILC: innate lymphoid cells; iNOS: inducible nitric oxide synthase; IL-1Ra: IL-1 receptor agonist; MHC-II: class II major histocompatibility complex; MPO: myeloperoxidase; PD-1: programmed cell death-1; PGE2: prostaglandin E2; TGF-β: transforming growth factor-β; Th: T helper; TNF-α: tumor necrosis factor-α. ↑: upregulation; ↓: downregulation.

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
