# Peer review of "Mechanisms behind the Immunoregulatory Dialogue between Mesenchymal Stem Cells and Th17 Cells"

_cells, 2020, doi:10.3390/cells9071660_

Round 1

Reviewer 1 Report

The review discusses the interaction between MSC and Th17 with particular focus on.

In my opinion the text is and in some way of interest for the reader.

Despite this, I might suggest some major  important modifications which would allow to improve the quality of the review

The Introduction and paragraph up to 1.3 seem a bit too similar and repetitive to what is already well acknowledged in literature regarding MSC. Normally we come across several reviews about definition of MSC and functions with the correspondent references (almost always the same references). Authors should rather focus on the immunosurveillance aspect.

It is unclear what angiogenesis, definition, and fibrotic effects of MSC have to do with the immune system if not properly associated to it. If authors desire to preserve these sections of the manuscript, it is desirable to discuss according to immune properties of MSC.

Accordingly, I would rather add other significant issues such as to discuss MSC behaviour in the context of cancer and not only from the anti-inflammatory point of view, also questioning when the biological interaction of MSC could really represent an advantage.

Finally, the review lacks novel biotechnologies and or molecular strategies to improve the interaction between MSC ad the immune system including Th17.

Author Response

Reviewer 1

The Introduction and paragraph up to 1.3 seem a bit too similar and repetitive to what is already well acknowledged in literature regarding MSC. Normally we come across several reviews about definition of MSC and functions with the correspondent references (almost always the same references). Authors should rather focus on the immunosurveillance aspect.

We fully agree with the reviewer comment and made significant modifications highlighted in yellow in the Introductionsection of the marked copy of revised manuscript. In that part, we focus more on the immunosurveillance aspect (lines 61-81).

It is unclear what angiogenesis, definition, and fibrotic effects of MSC have to do with the immune system if not properly associated to it. If authors desire to preserve these sections of the manuscript, it is desirable to discuss according to immune properties of MSC.

We fully agree with the reviewer comment and discuss according to immune properties of MSC (lines 47-81).

Accordingly, I would rather add other significant issues such as to discuss MSC behaviour in the context of cancer and not only from the anti-inflammatory point of view, also questioning when the biological interaction of MSC could really represent an advantage.

We fully agree with the reviewer comment and question when the biological interaction of MSC could represent an advantage (lines 61-81 and lines 147-156).

Finally, the review lacks novel biotechnologies and or molecular strategies to improve the interaction between MSC ad the immune system including Th17.

As requested by the reviewer, we added paragraphs on strategies aiming at improving the interaction between MSC and immune cells such as Th17 (Paragraph 4 lines 375-412).

Reviewer 2 Report

In this review article, Terrazza and colleagues tried to emphasize the role of mesenchymal stem cells in modulation of Th17 cells.

Although the topic is interesting, several issues limit scientific impact and significance of this article.

  1. Introduction is not directly related to the topic of this manuscript. Authors describe in detail functional characteristics of MSCs which are not related to the main topic (for example, MSC-dependent effects on angiogenesis, differentiation potential of MSC, etc…).
  2. In contrast, authors did not introduce readers with the role of Th17 cells in human pathology.
  3. In the section “Immunoregulatory properties of MSCs”, authors did not mention the effects of MSCs on Th2 cells, although MSC-based suppression of Th2 cells is well known.
  4. MSC-based effects on IL-23, IL-1, TGFb, IL-6 production in DCs, whichis crucially important for generation of Th17 cells is not described.
  5. Authors mentioned a lot of MSC-derived immunomodulatory factors (IL-10, PGE2, THGb, IDO, NO…) that play important role in MSC-based immunomodulation and MSC-dependent suppression of Th17 cells. However, authors did not dscribe the exact signaling pathways and precise molecular mechanisms which are involved in the effects of these molecules. This issue significantly limit the impact of this manuscript. Particularly when having in mind that authors submit their work to Cells, journal with IF>5.
  6. Authors design only one, very simple figure which does not describe several important findings discussed in this review.
  7. Several sentences and phrases are not properly written. For example, authors wrote: “It has been described that MSCs express chemokine receptors such CXCR3, CXCR4 or CCR5 [17,18] which promote migration to damaged tissues in pathological conditions, and ligands such CXCL12, CX2CL1, CXCL9, CXCL10 or CXCL11 [19,20], who promote the recruitment of different cells types” (lines 52-55). CXCR3, CXCR4 and CCR5 are not expressed only on MSCs and are also involved in homing of leucocytes to the inflamed tissues.
  8. Conclusion is not directly to the findings discussed in the main text.
  9. Moderate English language changes required.

Author Response

Reviewer 2

  1. Introduction is not directly related to the topic of this manuscript. Authors describe in detail functional characteristics of MSCs which are not related to the main topic (for example, MSC-dependent effects on angiogenesis, differentiation potential of MSC, etc…).

We fully agree with the reviewer comment and made significant modifications highlighted in yellow in the Introductionsection of the marked copy of revised manuscript. In that part, we focus more on the immunoregulatory properties of MSC (lines 61-81).

  1. In contrast, authors did not introduce readers with the role of Th17 cells in human pathology.

We agree with the reviewer comment and made the appropriate modifications by adding a paragraph on the role of Th17 cells in human pathology (lines 127-141 and Table 1).

  1. In the section “Immunoregulatory properties of MSCs”, authors did not mention the effects of MSCs on Th2 cells, although MSC-based suppression of Th2 cells is well known.

We agree with the reviewer comment and made the appropriate modifications (lines 119-121).

  1. MSC-based effects on IL-23, IL-1, TGFb, IL-6 production in DCs, which is crucially important for generation of Th17 cells is not described.

We agree with the reviewer comment and made the appropriate modifications (lines 325-329).

  1. Authors mentioned a lot of MSC-derived immunomodulatory factors (IL-10, PGE2, THGb, IDO, NO…) that play important role in MSC-based immunomodulation and MSC-dependent suppression of Th17 cells. However, authors did not describe the exact signaling pathways and precise molecular mechanisms which are involved in the effects of these molecules. This issue significantly limit the impact of this manuscript. Particularly when having in mind that authors submit their work to Cells, journal with IF>5.

We fully agree with the reviewer and accordingly describe the pathways and molecular mechanisms involved in MSC-dependent suppression of Th17 cells. All changes made to address that point are highlighted in yellow through the marked copy of revised manuscript.

  1. Authors design only one, very simple figure which does not describe several important findings discussed in this review.

We agree with the reviewer comment and made the appropriate modifications by adding more mechanistic details in the figure.

  1. Several sentences and phrases are not properly written. For example, authors wrote: “It has been described that MSCs express chemokine receptors such CXCR3, CXCR4 or CCR5 [17,18] which promote migration to damaged tissues in pathological conditions, and ligands such CXCL12, CX2CL1, CXCL9, CXCL10 or CXCL11 [19,20], who promote the recruitment of different cells types” (lines 52-55). CXCR3, CXCR4 and CCR5 are not expressed only on MSCs and are also involved in homing of leucocytes to the inflamed tissues.

We fully agree with the reviewer and improved the writing of this particular sentences (line 47-51) and others.

  1. Conclusion is not directly to the findings discussed in the main text.

As requested by the reviewer, we made some modifications in the Discussion.

  1. Moderate English language changes required.

We fully agree with the reviewer and improved the English language.

Round 2

Reviewer 1 Report

Authors still need to address the last point which is about the following: Finally, the review lacks novel biotechnologies and or molecular strategies to improve the interaction between MSC ad the immune system including Th17.

In other words authors need to address this point by highlighting novel molecular strategies that have been applied form a clinical stanpoint.

Authors have only reported molecular mechanisms, but what is important to understand is whether or not these molecular mechanisms have really been useful to proceed in clinical applicable strategies.  

Author Response

Reviewer 1

Authors still need to address the last point which is about the following: Finally, the review lacks novel biotechnologies and or molecular strategies to improve the interaction between MSC ad the immune system including Th17.

In other words authors need to address this point by highlighting novel molecular strategies that have been applied form a clinical stanpoint.

Authors have only reported molecular mechanisms, but what is important to understand is whether or not these molecular mechanisms have really been useful to proceed in clinical applicable strategies.  

We agree with the comment of the reviewer since at that stage preconditioned/pretreated MSC or their derivatives should be about to be used in clinic to treat patients with rheumatoid arthritis (RA). Since 2013, seven early phase trials have been conducted in RA. However, all of them used “naïve”/ unmodified MSC. Thus, to answer to the reviewer comments we added two paragraphs to develop where we are at the moment (lines 388-396 and lines 419-433).

English language

Of note, for this revised manuscript version, a qualified person fluent in English has provided a reliable edit to correct English language errors.

Reviewer 2 Report

Authors addressed all suggestions raised by me and significantly improved their work which is now, by my opinion, acceptable for publication.

Author Response

Reviewer 2

Authors addressed all suggestions raised by me and significantly improved their work which is now, by my opinion, acceptable for publication.
